# Frequency Balanced Datasets Lead to Better Language Models

**Rodolfo Zevallos**
Universitat Pompeu Fabra
Barcelona, Spain
rodolfojoel.zevallos@upf.edu

**Núria Bel**
Universitat Pompeu Fabra
Barcelona, Spain
nuria.bel@upf.edu

**Mireia Farrús**
Universitat de Barcelona
Barcelona, Spain
mfarrus@ub.edu

## Abstract

This paper reports on the experiments aimed to improve our understanding of the role of the amount of data required for training attention-based transformer language models. Specifically, we investigate the impact of reducing the immense amounts of required pre-training data through sampling strategies that identify and reduce high-frequency tokens as different studies have indicated that the existence of very high-frequency tokens in pre-training data might bias learning, causing undesired effects. In this light, we describe our sampling algorithm that iteratively assesses token frequencies and removes sentences that contain still high-frequency tokens, eventually delivering a balanced, linguistically correct dataset. We evaluate the results in terms of model perplexity and fine-tuning linguistic probing tasks, NLP downstream tasks as well as more semantic SuperGlue tasks. The results show that pre-training with the resulting balanced dataset allows reducing up to three times the pre-training data.

## 1 Introduction

Historically, using more training data has been considered a key to improving the performance of natural language processing (NLP) tools. The same seems to be true for transformer-based large language models (LLMs) such as BERT (Devlin et al., 2019) and RoBERTa (Liu et al., 2019) that exhibit their best performance when being trained with really massive quantities of texts. However, the question arises whether more training data is the only factor for improving results or whether specific characteristics of the provided texts have a particular impact on the learning efficiency of these powerful models. Recent research (Wei et al., 2021; Razeghi et al., 2022) has found evidence about the impact of the absolute frequency of pre-training data tokens in the prediction capacities of the model. Besides, high-frequency terms have

proved to be behind some phenomena related to the geometry of the representations causing problems for so called semantic tasks that rely on similarity assessments (Ethayarajh, 2019; Fuster-Baggetto and Fresno, 2022). In this paper, we report on our experiments to better understand the impact of token frequency in the pre-training data on language model (LM) quality, in terms of perplexity, and on the quality of the representations that these models provide for fine-tuning. Our ultimate goal is to study to what extent the quantity of pre-training data could be lessened, since this could be of great importance for training LM for low-resource languages (LRLs).

Texts are known to follow Zipf's law with some tokens occurring very frequently, some occurring with medium frequency, and many tokens that rarely occur forming a long tail. Different disciplines have addressed the problems of using long-tailed training data where some elements appear on most of the data but most of the elements are under-represented. Under-sampling and over-sampling techniques to directly adjust the number of examples by removing and adding data respectively have been proposed (Cui et al., 2019; Raunak et al., 2020). In this context, we propose an under-sampling algorithm whose objective is to balance token frequency by removing the sentences in which high-frequency tokens (and bigrams) occur, thus improving the estimation of model probabilities for low-frequency tokens. By ensuring that tokens have a balanced exposure during training, the model should become more capable of learning more diverse linguistic patterns, resulting in an improved model perplexity and in better representations for NLP tasks. Our experiments show that sampling the data to avoid a highly skewed token frequency distribution delivers results equivalent to pre-training the model with average three times more but unbalanced, raw data.

In this paper, we report on how our algorithm

processes the dataset, how it iteratively assesses token frequencies and removes sentences that contain still high-frequency tokens, eventually delivering a balanced, linguistically correct dataset. We evaluate the results in terms of model perplexity for English general and domain texts and for four other languages of different morphological complexities. Furthermore, we evaluate the approach in terms of the quality of the representations for fine-tuning to a range of tasks: linguistic probing tasks, NLP tasks and more semantic SuperGlue tasks. Our hypothesis was that if Fuster-Baggetto and Fresno (2022) analysis is correct, reducing frequency biases would also improve the results in semantic tasks.

In what follows, section 2 reviews the related work; section 3 describes the algorithm implemented, as well as its impact on the initial and the resulting datasets. Section 4 describes the experimental setup for training ten RoBERTa models with different data and in different languages and domains. Also in section 4, we describe the fine-tuning experiments for eleven different tasks that were meant to get thorough information about the impact of training the model with a balanced dataset, specially in semantic tasks. Section 5 describes the results of the experiments in terms of perplexity of the models and of F1-score of the different tasks. Results are analysed and discussed in section 6. Finally, section 7 is devoted to sum up the conclusions and contributions.

## 2  Related Work

A number of works have studied the impact of word frequency on different aspects of LLM and, in particular, on the quality of the delivered representations. Kassner et al. (2020) have studied BERT models and possible memorization based on token frequency, demonstrating that if a token appears fewer than 15 times, the model will disregard it, while a token that appears 100 times or more will be predicted more accurately. Zhou et al. (2022) demonstrated that high frequency words and low frequency words are represented differently by transformer LLM, in particular by BERT. Wei et al. (2021) found evidence that BERT models struggle to correctly predict a verb when it belongs to a set of word pairs (S-V), or bigrams, that appears less than 10 times or more than 100 times in the training corpus. Razeghi et al. (2022) also examined the strong impact of the frequency of terms in pre-

training data, although in a GPT model, and found a strong correlation of token frequencies with the resulting model performance.

Puccetti et al. (2022) provided evidence directly linking the presence of outlier dimensions (Kovaleva et al., 2021) with the frequency of tokens in the pre-training data. Outlier dimensions are those dimensions at which parameters with an unusually high magnitude —in terms of standard deviation–– are found consistently across the model layers. Puccetti et al. (2022) results suggest that outliers are due to the highly skewed token frequency distribution in the textual pre-training data. Moreover, Fuster-Baggetto and Fresno (2022) demonstrated that the non-fine-tuned BERT models contain token frequency biases that distort the embedding representation space. The distortion leads to the already observed poor performance on semantic tasks because of what Gao et al. (2019) already diagnosed as anisotropy in transformer LMs or the representation degeneration problem: the embeddings concentrate in a hypercone instead of occupying the whole space. Consistently, Ethayarajh (2019) found that randomly sampled words tend to be highly similar when measured by cosine similarity. Kovaleva et al. (2021) have also found evidence that different transformer-based LMs have similar behavior regarding poor semantic isometry, even when they differ in number of parameters, architecture or learning objective.

As for bias-removal techniques, our approach is different from Fuster-Baggetto and Fresno (2022), that preferred not to remove frequent tokens from sentences because it would be affecting the ability of the LM. Instead, they removed the embeddings after processing, before pooling like Jiang et al. (2022) and Yan et al. (2021). In our experiments, we remove the sentences with a high frequency token to guarantee that only grammatical sentences are processed. Our approach is, thus, in line with Samuel et al. (2023) findings that got a noticeable difference in downstream tasks performance when using the British National Corpus of 100M tokens, as a sample of a curated data set that proved to be enough to learn the required basic linguistic skills.

## 3  Frequency

In this section, we report on our algorithm that processes a raw text corpus and returns a balanced dataset. As explained in this section, which we set to a size 10M tokens —as we will explain later on

in more detail. The processing includes, on the one hand, calculating the number of occurrences of content tokens in the whole dataset and assessing to what extent they usually co-occur with other less frequent tokens, thus forming to be kept bigrams. On the other hand, frequency counts are updated after each sentence removal.

---

**Algorithm 1** Removing sentences from dataset

---

**Require:** $S, T_{max}, B_{min}, Freq, Bi$
**Ensure:** $S_{\text{balanced}}$
 1: $S_{\text{balanced}} \leftarrow S$
 2: $deleted\_phrases \leftarrow$ **True**
 3: **while** $deleted\_phrases$ **do**
 4:    $stop\_process \leftarrow$ **False**
 5:    **for** each $s_i$ in $S_{\text{balanced}}$ **do**
 6:       $W \leftarrow$ split_sentence($s_i$)
 7:       **if**    all $w_i$ in $W$, $Freq[w_i]$      $>$ $T_{max}$ and $Bi[(w_i,)] > B_{min}$ **then**
 8:          Remove $s_i$ from $S_{\text{balanced}}$
 9:          Update $w_i$ from $Freq$
10:          Update $(w_i,)$ from $Bi$
11:          $stop\_process \leftarrow$ **True**
12:       **end if**
13:    **end for**
14:    **if** $stop\_process ==$ **False then**
15:       $deleted\_phrases \leftarrow$ **False**
16:    **end if**
17: **end while**

---

### 3.1 Calculating token frequencies

To calculate the token frequencies of a dataset, we implemented the following steps:

1. Word split: We divided the dataset into individual tokens by using the space character (" ") as a delimiter. We opted for this easy approach as it was enough for the purpose of our experiments.

2. Stopword Removal: We eliminated stopwords from the dataset because they are mainly grammatical words that are among the most frequent tokens in any language. For our experiments, we used the Stopword function from NLTK[1] library, which contains stopwords list of German, French and Turkish. We got Quechua stopword from a Github repository.[2]

---

[1] https://www.nltk.org
[2] https://github.com/stopwords-iso/stopwords-iso

3. Counting Token Occurrences ($Freq$): We performed a count of how many times each token appeared in the dataset.

4. Bigram Counting ($Bi$): In addition to single tokens, we also considered bigrams, which are pairs of tokens appearing together. By counting the occurrences of bigrams, we wanted to take into account that very frequent tokens can consistently co-occur with other less frequent token that should be kept in the curated dataset.

In the steps described above, we obtain $Freq$ and $Bi$. $Freq$ is a list $\{(w_0, f_0), .., (w_i, f_i), .., (w_n, f_n)\}$, where $w_i$ is a content token and $f_i$ is the frequency of $w_i$. Similarly, $Bi$ is a list $\{((w_0,), fb_0), .., ((w_i,), fb_i), .., ((w_n,), fb_n)\}$, where $(w_i,)$ represents bigrams and $fb_i$ is the frequency of $(w_i,)$.

### 3.2 A frequency-algorithm for getting a balanced dataset

One of the most important steps in balancing a dataset is to setup an upper threshold for the occurrence of any token ($T_{max}$) or token and usual co-occurrences ($B_{min}$).

Taking into account the previous research, we decided to consider both token frequencies and co-occurrence frequencies in our algorithm. Furthermore, in order to adapt the algorithm to each dataset, we decided to set $T_{max}$ value as a relative value, calculated based on the average frequencies of the type tokens (TT), excluding outlier frequencies (OF). To remove the outlier frequencies, we used the "$outlier\_utils$"[3] library from Python. Our algorithm utilizes a $B_{min}$ of 10 and $T_{max}$ is obtained using the following equations:

$$\text{Freq}_{\text{avg}} = \frac{\sum_{n=1}^{Z} \text{Freq}_n}{Z} \qquad (1)$$

where:

$Z$ : Number of TT without OF.

$\text{Freq}_n$ : Frequency of token$_n$

$\text{Freq}_{\text{avg}}$ : Average frequency

$$T_{\max} = \begin{cases} \text{Freq}_{\text{avg}} & \text{if } \text{Freq}_{\text{avg}} < 100 \\ 100 & \text{if } \text{Freq}_{\text{avg}} \geq 100 \end{cases} \qquad (2)$$

---

[3] https://pypi.org/project/outlier_utils/#description

Algorithm 1 (removing sentences from dataset) is used to create a list of the sentences ($S_{\text{balanced}}$) that do not contain high frequency tokens. It takes as parameters all the sentences from the dataset ($S$), the list of token frequencies ($Freq$), the list of bigrams ($Bi$), $T_{max}$ and $B_{min}$.

To analyze each sentence ($s_i$) in $S_{\text{balanced}}$ where $\forall\ s_i \in S_{\text{balanced}} = \{s_0, ..., s_i, ..., s_n\}$, we split $s_i$ using (" ") as a delimiter, obtaining $W = \{w_0, ..., w_i, ..., w_n\}$, being each $w_i$ a token. Then, for each element of $W$ we obtain the values $Freq[w_i]$ and $Bi[(w_i, )]$. $Freq[w_i]$ and $Bi[(w_i, )]$ values are compared with $T_{max}$ and $B_{min}$, respectively. If in all $w_i$ $T_{max}$ and $B_{min}$ are lower than $Freq[w_i]$ and $Bi[(w_i, )]$, we remove $s_i$ from $S_{\text{balanced}}$; and we update the frequency counts in the lists $Freq$ and $Bi$. The process is repeated each time this condition is fulfilled, until the condition is not fulfilled in any $s_i$. The output of this process is $S_{\text{balanced}}$.

### 3.2.1 Dataset size details

In order to conduct the experiments, we followed Warstadt et al. (2020b), in which models created with 10M and 100M datasets where compared, and we used the algorithm to create balanced datasets of 10M tokens, except for the LRL Quechua, which we set at 1M tokens due to the lack of a large corpus. Table 1 shows the amount of data consumed for each dataset to reach 10M tokens using our algorithm.

| Dataset | Language | Size needed | Times |
|---------|----------|-------------|-------|
| Wikipedia | English | 36M | 3.6x |
| News | English | 29M | 2.9x |
| Reviews | English | 24M | 2.4x |
| BioMedic | English | 30M | 3.0x |
| CS | English | 19M | 1.9x |
| | German | 41M | 4.1x |
| OSCAR | French | 39M | 3.9x |
| | Turkish | 72M | 7.2x |
| Llamacha | Quechua | 6M | 6x |

Table 1: Details of raw datasets to result in a 10M balanced one in number of tokens and reduction times.

The amount of data required to reach 10M tokens varies significantly depending on the language and domain. In the case of Quechua, the size of the training data was 1M, and it required six times

more data to achieve the target. However, for most datasets, an average increase of four times was sufficient to reach 10M tokens, except for Turkish, which required approximately seven times more data.

## 4 Experimental Setup

In this section, we describe the setup of the different experiments. We trained ten RoBERTa models with different processed and non-processed datasets for English, French, German, Turkish data and measured their perplexity to assess the impact of balancing token frequency in pre-training data. We also applied the algorithm for assessing how it could be used in a LRLs scenario, such as the one of Quechua, with only a 6M tokens corpus that got reduced into only 1M. After that, we describe the fine-tuning of the English models for four different typical NLP tasks, for a linguistic probing classifier and for four different SuperGLUE tasks.

### 4.1 Language models

For training all the LMs, we followed the configuration outlined by Warstadt et al. (2020b) and specifically, we employed the hyperparameters from their Med-Small model, which consisted of 8 attention heads, a hidden size of 512, a feed-forward network dimension of 2048, and a total of 45 million parameters. In line with common practice in transformer-based LM development, we employed the BPE tokenizer (Sennrich et al., 2015), utilizing a vocabulary size of 52k tokens. Additionally, we adopted identical parameter values for dropout, attention dropout, and learning rate decay. For a comprehensive overview of these parameters, please refer to Table 8 in the appendix.

### 4.1.1 English Wikipedia corpus

English training data source is the Wikipedia corpus used by Devlin et al. (2019). It is relevant to note that this corpus has a total of 2.5 billion tokens but for our experiments we have only downloaded 50M tokens. We compared our roBERTa models trained with a balanced corpus of 10M tokens, obtained by using the frequency-algorithm, with the already available roberta-base models trained on 10M[4] and 100M[5] datasets.

---

[4] https://huggingface.co/nyu-mll/roberta-base-10M-2
[5] https://huggingface.co/nyu-mll/roberta-base-100M-2

### 4.1.2 English domain corpora

In addition to the experiments explained above, we trained RoBERTa models with data from different domains using our frequency-algorithm for sampling the data. The importance of using our frequency-algorithm on corpora other than Wikipedia lies in the need to observe how our algorithm behaves with domain-specific words and how they are distributed in a specialized corpus. We selected four specific domains for these experiments: news, product reviews, biomedicine, and computer science.

For the biomedicine and computer science domains, we utilized data obtained from Lo et al. (2019), with technical knowledge and specialized terminology in those fields. Regarding the news domain, we used data extracted from RealNews (Zellers et al., 2019). Finally, for the Amazon product reviews domain, we obtained training data from HuggingFace[6].

### 4.1.3 Other languages

In order to validate the robustness of the approach with other languages that are morphologically distinct from English —that is, with a different distribution of tokens and types—, we conducted experiments building models for German, French, Turkish, and Quechua. As mentioned in the introduction, we applied the algorithm to Quechua for assessing how it could be used in a LRL scenario with only a 6M tokens corpus.

The resources used for German, French, and Turkish were obtained from the OSCAR corpus (Ortiz Suárez et al., 2019). Each corpus has a different size. German has 21 billion tokens (Scheible et al., 2020), French has 32.7 billion tokens (Martin et al., 2020) and Turkish has 11.5 million documents (Toraman et al., 2022). For Quechua, we used the 6M corpus from Zevallos et al. (2022).

### 4.2 NLP downstream tasks

For English, we additionally fine-tuned a number of classifiers for traditional NLP tasks: Part-of-Speech tagging (POS), Dependency Labeling (DL) and Named Entity Labeling (NER). POS and NER task data were obtained from Weischedel et al. (2013)[7] and DL from Silveira et al. (2014).

We also addressed Relation Classification (REL) task, which is the task of predicting the relation that

holds in the real-world between two entities, as a sample of a more semantic NLP task. Different relations are taken from an inventory of symbolic relation types Cause-Effect; Content-Container, Instrument-Agency, etc. The data were taken from Tenney et al. (2019).

All fine-tuning models used same hyperparameters. For a comprehensive overview of these parameters, please see the appendix.

### 4.3 BLiMP tasks

We utilized the BLiMP testset (Warstadt et al., 2020a) to evaluate the quality of representations regarding individual grammatical phenomena in English. BLiMP is a collection of 67 tasks, each of which contains 1000 minimal pairs of sentences that highlight specific morphological, syntactic, or semantic phenomena. These BLiMP minimal pairs consist of two sentences that differ in a single edit but contrast in their grammatical acceptability. The purpose of BLiMP is to conduct unsupervised evaluation of LMs through a forced-choice acceptability judgment task. A LM is considered to classify a minimal pair correctly if it assigns a higher probability to the acceptable sentence. Thus, BLiMP provides a valuable tool for measuring and comparing the performance of models. For evaluation, we followed the MLM scoring method proposed by Salazar et al. (2020).

### 4.4 SuperGLUE tasks

SuperGLUE is a benchmark set consisting of eight classification-based language understanding tasks Wang et al. (2019). We evaluated English LMs fine-tuning five SuperGLUE tasks in order to see the impact of training with a balanced dataset in tasks that are considered to involve understanding such as Boolean Questions (BoolQ) (Clark et al., 2019), Commitment Bank (CB) (De Marneffe et al., 2019), Choice of Plausible Alternatives (CPA) (Gordon et al., 2012), Recognizing Textual Entailment (RTE) (Wang et al., 2019) and Word in Context (WiC) (Pilehvar and Camacho-Collados, 2019). Our interest was whether reducing frequency biases would also improve the results in semantic tasks. Description of the tasks and the hyperparameter search range used for each task are described in the appendices.

---

[6]https://huggingface.co/datasets/amazon_us_reviews

[7]https://catalog.ldc.upenn.edu/LDC2013T19

| English Wikipedia | |
|---|---|
| **Model** | **Perplexity** |
| roberta-base-100M | **4.61** |
| roberta-base-10M | 10.78 |
| roberta-freq-10M | **4.93** |

Table 2: Model perplexity of language models trained with raw data and balanced word token frequency. The word "freq" in the model name refers to the use frequency-balanced data after applying our algorithm.

## 5 Results

We now describe the results of the different experiments. In all of them, the use of a balanced dataset for training the LMs equals or improves its quality in terms of perplexity of the LM or in terms of accuracy in fine-tuned tasks[8].

### 5.1 Language models and pre-training data

#### 5.1.1 English Wikipedia

To assess the results, we evaluated the impact of balancing the pre-training datasets by comparing the perplexity of the model obtained with the perplexity of models trained with the same amount but raw data.

Table 2 shows the perplexity of "roberta-base-10M" and "roberta-base-100M" as obtained with the models from Zhang et al. (2020), as well as the model trained with the frequency balanced corpus. The perplexity is reduced in almost 6 points, from 10.78 to 4.93, getting very close to 4.61 perplexity of the model when trained with 100M words corpus of raw data.

#### 5.1.2 Domains

We also evaluated different domain-specific training sets collected from domains such as news, reviews, biomedical, and computer science texts. As shown in Table 3, the reduction of model perplexity is clear in all the domains, with 6 points of reduction in average, which supports the robustness and consistency of the impact of balancing word token frequency.

#### 5.1.3 Other languages

Like with the roBERTa model of English, we evaluated the perplexity of models trained with corpora

[8]Improvements are in all cases statistically significant at p < 0.05 as assessed with a sign test

| English Domains | | |
|---|---|---|
| **Domain** | **roberta-base** | **roberta-freq** |
| News | 19.42 | 13.59 |
| Reviews | 25.17 | 18.26 |
| BioMedic | 17.31 | 10.15 |
| Computer Science | 15.48 | 8.63 |

Table 3: Model perplexity of language models trained with different domains and applying the frequency-algorithm to get 10M tokens training data. These models were trained from scratch for this experiment.

from different languages to see the impact of using our frequency algorithm. As shown in Table 4, German achieves a reduction in perplexity of more than 7 points using the balanced corpus (roberta-freq). Similarly, French, Turkish and Quechua reduce perplexity by 8.26, 8.65 and 62.75 points respectively, confirming that more morphologically complex languages, with more word types due to inflection, also greatly benefit of a balanced pre-training dataset.

| Other Languages | | |
|---|---|---|
| **Language** | **roberta-base** | **roberta-freq** |
| German | 17.05 | 9.76 |
| French | 13.81 | 10.88 |
| Turkish | 35.18 | 26.53 |
| Low-Resource Language | | |
| **Language** | **roberta-base** | **roberta-freq** |
| Quechua | 358.47 | 295.72 |

Table 4: Model perplexity of language models trained with different languages when applying the frequency-algorithm. All models are trained with 10M tokens except for Quechua, which is trained with only 1M tokens. These models were trained from scratch for this experiment.

### 5.2 NLP downstream tasks

As can be seen in Table 5, the use of our frequency reduction algorithm to process the training dataset leads to improvements in the four tasks: POS, NER, REL and Dependencies. The "roberta-freq-10M" model increases performance compared to "roberta-base-10M", with better results for tasks that were more semantic such as REL. Moreover, note that the results in terms of F1-score are the same for

| NLP Task | | | | |
|---|---|---|---|---|
| **Model** | **Part-of-Speech** | **NER** | **REL** | **Dependencies** |
| roberta-base-100M | **0.98** | **0.97** | **0.76** | **0.92** |
| roberta-base-10M | 0.96 | 0.95 | 0.67 | 0.88 |
| roberta-freq-10M | 0.97 | **0.97** | **0.76** | **0.92** |

Table 5: NLP task results in F1-score for the different language models trained according to: training size and frequency reduction technique.

| BLiMP | | | |
|---|---|---|---|
| **Task** | **roberta-base-10M** | **roberta-freq-10M** | **roberta-base-100M** |
| ANA. AGR. | 91.1 | 96.7 | **97.2** |
| ARG. STR | 70.1 | 78.9 | **79.1** |
| Binding | 71.6 | 75.2 | **75.4** |
| CTRL RAIS | 70.7 | **79.7** | 79.6 |
| D-N ARG. | 91.6 | 93.1 | **94.5** |
| Ellipsis | 86.0 | 90.8 | **91.6** |
| Filler GAP | 67.3 | **79.1** | 78.8 |
| Irregular | 84.3 | 92.5 | **92.7** |
| Island | 53.6 | 61.3 | **63.0** |
| NPI | 75.6 | **77.3** | 77.2 |
| Quantifiers | 58.6 | 62.7 | **64.7** |
| S-V ARG. | 77.0 | 86.8 | **87.5** |

Table 6: BLiMP results for the 10M and 100M roberta-base models and the roberta model using our 10M frequency-algorithm.

almost all cases that the scores obtained with the "roberta-base-100M" model.

### 5.3 BLiMP Tasks

The experiments conducted with the BLiMP dataset reveal improvements regarding the reference models used: "roberta-base-10M" (see Table 6) in most of the tasks. In the ANA.AGR task, a score of 96.7 is achieved with the "roberta-freq-10M" model, surpassing "roberta-base-10M" by 5.6 and coming close to the "roberta-base-100M" model with 97.2. In the CTRL.RAIS task, the "roberta-freq-10M" model achieves a score of 79.7, surpassing "roberta-base-10M" by 9.0 and "roberta-base-100M" by 0.1. Similarly, in the NPI task, the "roberta-freq-10M" model demonstrates superiority over the "roberta-base-10M" and "roberta-base-100M" models by 1.7 and 0.1 respectively. As for the Filter.GAP task, the "roberta-freq" model outperformes both base models, achieving a score of 79.1. Lastly, the "roberta-freq-10M" model surpasses the "roberta-base-10M" model in all tasks.

### 5.4 SuperGLUE tasks

The experiments conducted on the SuperGLUE dataset reveal important findings regarding the performance of the evaluated models. The "roberta-freq-10M" model achieves comparable or slightly better performance than the baseline models in most of the tasks evaluated in SuperGLUE.

As can be seen in Table 7, the most highlighting task where the "roberta-freq-10M" equals "roberta-base-100M" is WiC, achieving a score of 0.70. Moreover, the "roberta-freq-10M" overcomes the "roberta-base-10M" model in all tasks, with an average of 0.05 points.

### 6 Discussion

Since word frequency shows a highly skewed distribution in raw texts, when using a balanced dataset (especially when pre-training with a small 10M corpus), the potential biases due to highly-frequent tokens should be moderated, and the system should be able to perform a better computation of the prob-

| SuperGLUE | | | |
|---|---|---|---|
| **Task** | **roberta-base-10M** | **roberta-freq-10M** | **roberta-base-100M** |
| BoolQ | 0.68 | 0.73 | **0.74** |
| CB | 0.81 | 0.86 | **0.89** |
| Copa | 0.57 | 0.60 | **0.61** |
| RTE | 0.57 | 0.62 | **0.64** |
| WiC | 0.65 | **0.70** | **0.70** |

Table 7: SuperGlue results for the 10M and 100M roberta-base models and the roberta model using our 10M frequency-algorithm.

abilities of medium and rare words.

The results of our experiments consistently show that perplexity in all our experiments improves when working with a balanced corpus. In particular, we have seen that using balanced datasets for modeling morphological complex languages with more number of types due to inflection like German, French, Turkish and Quechua also reduces model perplexity, confirming the robustness of the approach with different probability distributions.

The improvement of rare tokens modeling after eliminating high-frequency tokens is clearly shown in the BLIMP results, which show a better performance for a model trained with a balanced 10M corpus than a model trained with a 100M words corpus for tasks related to medium or rare words like Negative Polarity Items (NPI) like 'ever' or Control/Raising verbs like 'oblige' or 'promise'. Note that negative sentences are a minority in English written texts ranging from less than 10% to 32% of the sentences in different reference corpora (Jiménez-Zafra et al., 2020). Also note that Filler-Gap phenomena mostly occur in question sentences that are less frequent in any corpus of English too (see, for instance, Liu et al. (2022)).

Moreover, since high-frequency words seem to be responsible for representations with outlier components distorting the geometry of the created hyperspace, semantic fine-tuned tasks should improve more than others that do not rely on semantic similarity. For instance, we see an important improvement in the Relation Classification task (REL), which predicts the relation that holds between two entities, and also an improvement that equals the results of pre-training with 100M in the task of Word in Context (WiC) —a word sense disambiguation task. Therefore, our results support Fuster-Baggetto and Fresno (2022) because we found that the lack of frequency bias improves the quality of

the semantic representation and makes the system perform significantly better especially in semantic tasks.

## 7 Conclusions

In the current paper, we reported on an algorithm that processes a corpus assessing token frequencies to remove sentences that contain high-frequency tokens, eventually delivering a balanced, linguistically correct data set. We have shown that training with a balanced corpus improves the quality of the model, as shown in a significant reduction in perplexity for English general and domain texts and for five other languages of different morphological complexities. Furthermore, the results also show the quality of the representations for fine-tuning tasks: linguistic probing tasks, NLP tasks and SuperGlue tasks also improve the results obtained with a non-balanced, raw dataset, in many cases delivering results that equals the results obtained with models trained with 100M tokens of raw text. In particular, such better results in semantic tasks confirm previous researches that signalled high-frequency tokens distorting the semantic space created by the models.

Summing up, the contributions of our research are the algorithm for balancing textual data, the different balanced corpora that are available with the code at ANONYMIZED, and further evidence about the impact of high-frequency tokens for training transformed-based LMs. Finally, we think that our findings will be of great interest for LRLs, since we have defined the characteristics of the data to be gathered or created so that, using less data, the created models can achieve the best results.

## Limitations

We have limited ourselves to studying only four domains other than Wikipedia because the majority

of other types of domains are not publicly accessible or do not have the minimum required data for our experiments.

On the other hand, we have chosen to use Quechua as the language of low-resource settings for being one of the few languages that meets the necessary requirements for our experiments. Quechua is a morphologically complex language (agglutinative), and it has a dataset of 6M tokens for training LMs.

## Ethics Statement

The datasets used in this paper for the training and evaluations of the pre-trained models, and fine-tuned models have been extracted from various previous articles and open-access repositories. Therefore, we abide by the ethical rules by citing the original authors of each dataset. In addition, we encourage authors who use the resources in this article to cite the original sources. Finally, it is important to mention that one of the authors is an active member of various initiatives and non-governmental organizations (NGO) aimed at the preservation and revitalization of minority languages. As a result, this provides a deeper understanding of the issues and concerns surrounding minority languages.

## Acknowledgements

This research was partially funded by the project LUTEST, Project PID2019-104512GB-I00, Ministerio de Ciencia, Innovación y Universidades and Agencia Estatal de Investigación (Spain). The first author has been supported by a FI grant of the Catalan Funding Agency for Research and Universities (AGAUR).

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

## A  Appendices

### A.1  Language Models details

To train all LMs, we followed the choices by Warstadt et al. (2020b) for their RoBERTa Med-Small model with 45M parameters. Table 8 show the hyperparameters used in our experiments.

| Description | Value |
|---|---|
| Number of Layers | 6 |
| Hidden size | 512 |
| Feed-forward network dimension | 2048 |
| Attention heads | 8 |
| Dropout | 0.1 |
| Attention dropout | 0.1 |
| Max Steps | 10K |
| Learning rate decrease | 5E-4 |
| Batch Size | 512 |
| Number of parameters | 45M |

Table 8: Configuration used for all RoBERTa model trainings

We ran all training on 5 server equipped with an Intel Xeon E5-2650 v4 CPU (12 cores, 2.2GHz 30MB Cache 2400MHz 105W) and a Gigabyte Geforce GTX 1080 Ti TURBO 11.72GB GPU. We trained each model for 10k steps. The training time was over 4 days for models of 10M and 15 hours for models of 1M. The entire LMs creation experiment took approximately 8 days.

### A.2  NLP downstream tasks details

From the pretrained RoBERTa models, and still following Zevallos and Bel (2023), we generated representations of the token span and trained classifiers that predict whether a given label correctly describes the input span for NER, POS, Dependency Labeling (DL) and Relation Classification (REL).

In order to obtain the best and validated results in all tasks, we performed a 10-fold macro-F1 score cross-validation. We set a batch size of 16 and a learning rate of 2E-5 in all our fine-tuning models. Furthermore, for evaluating the four tasks, we used macro-F1 score.

On the other hand, we have used 1 server equipped with an Intel Xeon E5-2650 v4 CPU (12 cores, 2.2GHz, 30MB Cache, 2400MHz, 105W) and a Gigabyte GeForce GTX 1080 Ti TURBO 11.72GB GPU to train all these models. The training time was approximately 5 hours per task and

model. In total, we trained 12 models over a period of approximately 4 days.

### A.3  BLiMP and SuperGlue details

The tasks of BliMP and SuperGlue are described in Table 9 and Table 10 respectively.

| Task | Fullname | Description |
|------|----------|-------------|
| ANA. AGR. | Anaphor agreement | the requirement that reflexive pronouns like himself (a.k.a. anaphora) agree with their antecedents in person, number, gender, and animacy. |
| ARG. STR | Argument structure | The ability of different verbs to appear with different types of arguments. For instance, different verbs can appear with a direct object, participate in the causative alternation, or take an inanimate argument. |
| Binding | Binding | The structural relationship between a pronoun and its antecedent. |
| CTRL RAIS | Control/raising | Syntactic and semantic differences between various types of predicates that embed an infinitival VP. This includes control, raising, and tough-movement predicates. |
| D-N ARG. | Determiner-noun agreement | Number agreement between demonstrative determiners (e.g., this/these) and the associated noun. |
| Ellipsis | Ellipsis | The possibility of omitting expressions from a sentence. Because this is difficult to illustrate with sentences of equal length, our paradigms cover only special cases of noun phrase ellipsis that meet this constraint. |
| Filter GAP | Filler-gap | Dependencies arising from phrasal movement in, for example, wh-questions. |
| Irregular | Irregular forms | Irregular morphology on English past participles (e.g., broken). We are unable to evaluate models on nonexistent forms like *breaked because such forms are out of the vocabulary for some LMs. |
| Island | Island effects | Restrictions on syntactic environments where the gap in a filler-gap dependency may occur. |
| NPI | NPI licensing | Restrictions on the distribution of negative polarity items like any and ever limited to, for example, the scope of negation and only. |
| Quantifiers | Quantifiers | Restrictions on the distribution of quantifiers. We cover two such restrictions: superlative quantifiers (e.g., at least) cannot embed under negation, and definite quantifiers and determiners cannot be subjects in existential-there constructions. |
| S-V ARG. | Subject-verb agreement | Subjects and present tense verbs must agree in number. |

Table 9: Description of each BLiMP task used in our experiments.

| Task | Fullname | Description |
|------|----------|-------------|
| BoolQ | Boolean Questions (Clark et al., 2019) | BQ, is a QA task where each example consists of a short passage and a yes/no question about the passage. The questions were provided by users of Google SE, and afterwards paired with a paragraph from a Wikipedia article containing the answer. |
| CB | CommitmentBank (De Marneffe et al., 2019) | CB is a corpus of short texts in which at least one sentence contains an embedded clause. The embedded clause is annotated with the degree of commitment expressed by the author of the sentence, and the task is to predict it. |
| COPA | Choice of Plausible Alternatives (Gordon et al., 2012) | COPA is a causal reasoning task that chooses the cause or effect of a premise sentence from two possible choices. |
| RTE | Recognizing Textual Entailment (Wang et al., 2019) | RTE task is about text entailment derived from the Natural Language Inference (NLI) dataset in which given a premise sentence and a hypothesis sentence, the task is to predict whether the premise entails the hypothesis or not. |
| WiC | Word in Context (Pilehvar and Camacho-Collados, 2019) | WiC is a kind of disambiguation task. Given two text snippets and a polysemous word that appears in both sentence, the task is to decide whether the polysemous word is used with the same sense in both sentences. |

Table 10: Description of each SuperGLUE task used in our experiments.