# OpenReview forum: "Frequency Balanced Datasets Lead to Better Language Models"
_EMNLP/2023/Conference — EMNLP 2023 Findings_

### Official Review · Reviewer_zg87 · 2023-08-03

**Soundness:** 3

**Excitement:**

3: Ambivalent: It has merits (e.g., it reports state-of-the-art results, the idea is nice), but there are key weaknesses (e.g., it describes incremental work), and it can significantly benefit from another round of revision. However, I won't object to accepting it if my co-reviewers champion it.

**Missing References:**

Maybe refer to this https://arxiv.org/pdf/2102.09690.pdf and compare some works on the "common token bias" in the Related Works section of this paper.

**Paper Topic And Main Contributions:**


Summary: This paper presents a method to select/sample data for language modelling (LM) by removing sentences with sentences with high frequency tokens, hypothesizing that resulting corpus will be a more balance corpus.

Highlights:

- [Competitive Results with less data] On the intrinsic LM metric is the achievement of comparable LM perplexity with 10x less data on the English wikipedia
- [SOTA Results] Across various domains, for English, models trained with proposed data filtering method achieved lower perplexity
- [SOTA Results] Comparatively, for non-English other languages chosen in the experiments, models trained with filtered data using the proposed algorithm achieved lower perplexity
- On downstream superglue tasks, their proposed data filtering method improved model’s performance with comparable corpus size but the model trained on 10x corpus size outperforms the model trained on filtered data
- On downstream blimp tasks, their proposed data filtering method achieved competitive performance as compared to models trained on 10x corpus size


Strength:

- Simple proposal to reduce corpus size to achieve LM perplexity compared to models trained on 10x data size
- Achieved competitive results in downstream tasks that test the grammatical integrity of the language model (blimp)

Weakness:

- While the proposed method is compared to model trained with no filtering, it would be good to at least see the comparison of other filtering methods, e.g. simple perplexity filtering (filter low/high perplexity items based on n-gram / other pre-tained language model).
- Downstream tasks were all English centric, it’s hard to determine the efficacy of the proposed method in non-English language esp. when the perplexity improvements don’t significantly improve downstream tasks for English


Suggestions:
- Use bold in tables, avoid using red in tables (or generally colours on paper to accommodate colour-blind readers)
- To improve soundness/technical strength score, a post-hoc (after tokenised training) analysis should be done to see coverage of the tokens for low-frequency work, e.g. section 2 from https://aclanthology.org/2023.americasnlp-1.18
- To improve your “excitement” score, capitalise on the chosen language families Quechua and Turkish (both agglutinative), if you have a specific downstream task that test generation/labelling/annotation for agglutination that’ll make the paper very strong accept (even if the model don’t perform well, you’ve explored the limitation of the model)
- To make it a paper even more "exciting", add machine translation to the mix of downstream task, even though the model is trained monolingual, fine-tuning the model with a small corpus in the languages available in the pre-trained LM would show how well a user can capitalise on the “balancedness” of the corpus the paper suggested to create better/worse cross lingual information (shown by proxy with MT scores)

**Questions For The Authors:**

Questions:
- Other than stop words and split by whitespace are punctuations removed?
- Is the perplexity scores calculated based on 100M corpus even though it’s trained on 10M or is the model trained on 10M measured on the 10M?

**Reasons To Accept:**

Simple approach, clear writing. A marginal accept, 3.5 out of 5 score for accepting the paper. More details, see strength above.

**Reasons To Reject:**

Fundamentally, the paper is sound but a little extra work (in suggestions above) would make the paper a lot better. Would prefer to read the paper if the suggestions are incorporated in revision / future re-work of the paper. See weakness above for details.

**Reproducibility:**

4: Could mostly reproduce the results, but there may be some variation because of sample variance or minor variations in their interpretation of the protocol or method.

**Reviewer Confidence:**

3: Pretty sure, but there's a chance I missed something. Although I have a good feel for this area in general, I did not carefully check the paper's details, e.g., the math, experimental design, or novelty.

---

> ### Author Rebuttal · Authors · 2023-08-28
>
> Dear R4,
>
> Thanks for your encouraging comments. Before answering your questions we would like, for clarification, comment on the weaknesses you mention in the review.
>
> We are also very curious about how to explain that perplexity improvements do not improve the results in downstream tasks for English, and to get evidence of these tasks for other languages. This is one of our future lines of research, and, if accepted, we will highlight it in the conclusions and limitations.
>
> We have mentioned other filtering methods that address, as we do, high-frequent words. Fuster-Baggetto & Fresno (2022) are the only ones that evaluate an approach to get rid of frequent words, but not balancing the data-set: just removing embeddings, mostly of very frequent stop words. To the best of our knowledge, there are no other experiments comparable to ours. Therefore, it is not possible to provide a comparison with other filtering algorithms in this case. Also note that our results support other’s results related to balanced datasets: Samuel et al (2023) found that a noticeable difference in downstream task performance was achieve used the 100M BNC corpus, that it is not balanced with respect high frequency words, but it is a curated corpus.
>
>
> As for your questions:
>
> 1. Other than stop words and split by whitespace are punctuations removed?
>
> All the models in our experiments have been trained with their respective punctuation marks. However, in the frequency count phase, we removed the punctuation marks, since they are not relevant both in frequency term and co-occurrence.
>
> 2. Is the perplexity scores calculated based on 100M corpus even though it’s trained on 10M or is the model trained on 10M measured on the 10M?
>
> The perplexity of each model is based on the size of the training set. For instance, if we train a model of 10M in English, the dev set for the perplexity is 1M. Both datasets (train and dev) are taken from the 100M of the English corpus.

---

### Official Review · Reviewer_VqbC · 2023-08-05

**Typos Grammar Style And Presentation Improvements:** Suggest to rename the misleading toke…
**Soundness:** 3

**Excitement:**

3: Ambivalent: It has merits (e.g., it reports state-of-the-art results, the idea is nice), but there are key weaknesses (e.g., it describes incremental work), and it can significantly benefit from another round of revision. However, I won't object to accepting it if my co-reviewers champion it.

**Paper Topic And Main Contributions:**

The paper presents the analysis of how the balanced datasets could help to achieve similar results on downstream tasks with smaller dataset sizes. In line with previous research [1], the authors show that smaller datasets are enough because they take into consideration most syntactic and semantic features. The paper introduces the method for cutting the sentences with the most frequent words (including bigrams) and compares the obtained corpus of 10M words on different downstream tasks: English Wiki, PoS, NER, BLiMP, several tasks from SuperGLUE, and others. Also, several multilingual comparisons are done for German, French, Turkish, and low-resource Quechua languages. Almost for all benchmarks the obtained 10M-freq dataset overperforms 10M dataset.  All experiments are conducted with RoBERTa backbone.

The paper is well-written and effectively conveys the main idea. It suggests a quite simple and effective method to reduce the dataset size. With the suggested approach it seems reasonable to achieve better downstream results with a smaller dataset.

However, I have a few unclear points, that I want to discuss. First, it seems that the results for roberta-base-100M and roberta-base-10M are taken from previous works (i.e. BLiMP, English Wikipedia), and different sampling strategies could lead to different results. Second. I am not sure that a direct compare 10M and 10m-freq is the correct setup. As far as 10M-freq usually is obtained from 3x-4x larger number of data I suggest providing the comparison of 10M-freq with full data, which was used to obtain smaller corpus.

Strengths:

1) The suggested approach is quite easy to implement and can lead to better results with smaller datasets.

Weaknesses:

1)  Need to evaluate the obtained results with full datasets used for obtaining 10M-freq. Not clear whether the baseline comparison is correct (whether were sampled same data)
2) The suggested approach is quite straight-forward and there is no specification in what setups it could be beneficial

[1] [When Do You Need Billions of Words of Pretraining Data?](https://aclanthology.org/2021.acl-long.90) (Zhang et al., ACL-IJCNLP 2021)

**Questions For The Authors:**

A. How does align the results for roberta-base-100M and roberta-base-10M, that was taken with previous works (i.e. BLiMP, English Wikipedia)? It seems the different sampling strategies could lead to different results.
B. What are the results of training on the full dataset used for obtaining the 10M-freq dataset? I.e. on 36M-wiki and others. For competitive comparison, I suggest including it.

**Reasons To Accept:**

1) The suggested approach is quite easy to implement and can lead to better results with smaller datasets.

**Reasons To Reject:**

1)  Need to evaluate the obtained results with full datasets used for obtaining 10M-freq. Not clear whether the baseline comparison is correct (whether were sampled same data)
2) The suggested approach is quite straight-forward and there is no specification in what setups it could be beneficial

**Reproducibility:**

5: Could easily reproduce the results.

**Reviewer Confidence:**

4: Quite sure. I tried to check the important points carefully. It's unlikely, though conceivable, that I missed something that should affect my ratings.

---

> ### Author Rebuttal · Authors · 2023-08-28
>
> Dear R3,
>
> We really appreciate your detailed comments and contributions to improve our paper.  We would like you to note that, trying to clarify your following comment “It suggests a quite simple and effective method to reduce the dataset size. With the suggested approach it seems reasonable to achieve better downstream results with a smaller dataset.”. Our aim was to understand  the impact of high-frequent words, not of the dataset size, although it is a side-effect. Indeed, there are other algorithms that just try to reduce the size, but not paying attention to what in a corpus is less efficient, which after our results are high-frequent words. Our results demonstrate that for equal size corpora, the one with balanced frequency delivers better results.
>
> Although it is not one of your questions, we would also like to clarify what is the setup that could benefit from the findings of our research as you expressed doubts about it. We think that for low-resource languages (or for domains) that need to plan how to develop larger pre-training datasets it could be of paramount importance not to try to wildly collect as much data as possible, but to design a balanced dataset.
>
> As for your questions:
>
> 1. How does align the results for roberta-base-100M and roberta-base-10M, that was taken with previous works (i.e. BLiMP, English Wikipedia)? It seems the different sampling strategies could lead to different results.
>
> We took results for English, and when possible, from previous publications to save resources. However, we would like to note that we are working with subsets, like Wardstat et al. 2020. We would also like to draw your attention to the fact that we are interested in getting evidence about the impact of having a balanced pre-training dataset vs. raw dataset of the same amount of data. This is why we keep comparing with other 10M corpora, and then with a 100M to see the actual impact of the improvement.
>
> 2. What are the results of training on the full dataset used for obtaining the 10M-freq dataset? I.e. on 36M-wiki and others. For competitive comparison, I suggest including it.
>
> We have not compared the balance dataset with the raw-source dataset because the main focus of the paper was to demonstrate that high frequency words harm the modeling, and that balancing word frequency benefits the final results for two equal size datasets. But we agree that the comparison you suggest would also be interesting. We will include it if the paper is accepted, or anycase in the future versions as we plan to extend our experiments and send the results to a journal paper where all these details could be added

---

### Official Review · Reviewer_oSA3 · 2023-08-08

**Typos Grammar Style And Presentation Improvements:** 3.1 first time language is mentioned …
**Soundness:** 4

**Excitement:**

4: Strong: This paper deepens the understanding of some phenomenon or lowers the barriers to an existing research direction.

**Paper Topic And Main Contributions:**

I've updated my score. However, I really think that the authors need to include the information about the overlap in vocab terms in the paper. It makes a big difference to a reader on comparing perplexities. I would discount the paper without it.

```````````````````````

This paper looks at what happens when data is preprocessed to remove sentences that have high frequency terms. The overall idea is to prevent a masked language model (Roberta) from focusing too much on high frequency tokens and instead to focus on low-frequency. Experiments were run on a large set of tasks – though mostly classification based and perplexity. Experiments were run on a large set of tasks – though mostly classification based and perplexity. Overall, the authors look at 5 languages including English and showed that performance improved over the baseline 10M, while approaching the 100M.

**Questions For The Authors:**

- How is perplexity calculated? Is it on a held-out dev set?
- What's the vocabulary? It says 52k, but is it the same vocabulary for all models?
- What happens if you apply your method to 100M? Would it improve over the 100M baseline?

**Reasons To Accept:**

This is an interesting idea that attempts to deal with power-law issues that may imply we as a field are brute-forcing in a non-intelligent way. The results seem promising - though I have some questions for the authors.

**Reasons To Reject:**

- I'm a bit unclear on the experimental setup for the perplexity experiments. Are the vocabularies the same? This can impact perplexity calculations? Was a held out dev set used? 5.1.1 doesn't appear to be, but maybe 5.1.2 (different domains are)?
- Generalization beyond this model architecture and data size (though the experiments are rather thorough)

**Reproducibility:**

4: Could mostly reproduce the results, but there may be some variation because of sample variance or minor variations in their interpretation of the protocol or method.

**Reviewer Confidence:**

4: Quite sure. I tried to check the important points carefully. It's unlikely, though conceivable, that I missed something that should affect my ratings.

---

> ### Author Rebuttal · Authors · 2023-08-28
>
> Dear R2,
>
> Thanks a lot for your positive comments. Indeed we are interested in how to prevent a masked language model (Roberta) from focusing too much on high frequency tokens and ignoring low-frequency ones. As you suggest, it seems to us that brute force could solve many problems by raising the number of samples for all tokens that would no longer be ignored. However, this approach seems unsustainable, as there are no massive amounts of data for all languages, or  even all domains.
>
> As for your questions:
>
> 1. How is perplexity calculated? Is it on a held-out dev set?
>
> We used the code provided in https://github.com/facebookresearch/fairseq to compute perplexity, in the same way as Liu et al (2019). The dev set is a held out 10% of the dataset.
>
> 2. What's the vocabulary? It says 52k, but is it the same vocabulary for all models?
>
> All the experiments used BPE as tokenizer, and the vocabulary size was set at 52K in all cases. It is true that the perplexity could be affected if the different models have different vocabularies. After your comment, we have compared the words in the vocabularies of the 3 different English models used for the experiments. Percentage of coincidences is in this table. As you can see, the percentage of coincidences is almost 100% for the two models working with a corpus of 10M.
>
> _Table 1: Comparison of vocabulary tokens that share between datasets._
>
> | Datasets     | **10M** | **10M-Freq** | **100M** |
> |--------------|:-------:|:------------:|:--------:|
> | **10M**      |    X    |     99.6%    |   97.1%  |
> | **10M-Freq** |  99.6%  |       X      |   97.8%  |
> | **100M**     |  97.1%  |     97.8%    |     X    |
>
> 3. What happens if you apply your method to 100M? Would it improve over the 100M baseline?
>
> For computational reasons, and with an approach to work with low-resourced languages, we focussed on our experiments up to 10M words. In the future, however, we plan to extend the parameters of the experiments, and to analyze to what extent the number of samples of medium to low-frequency words, that would be more in a larger corpus, could affect the balancing effects of the algorithm, and therefore, the performance of the models.

---

### Official Review · Reviewer_gnCA · 2023-08-11

**Soundness:** 4

**Excitement:**

3: Ambivalent: It has merits (e.g., it reports state-of-the-art results, the idea is nice), but there are key weaknesses (e.g., it describes incremental work), and it can significantly benefit from another round of revision. However, I won't object to accepting it if my co-reviewers champion it.

**Paper Topic And Main Contributions:**

This paper proposes to reduce the amount of data needed to achieve good language modeling perplexity and downstream task performance, especially that on semantic tasks, by removing sentences consisting entirely of high frequency tokens.

List of contributions: new data resources, NLP engineering experiment, publicly available software (upon publication).

**Questions For The Authors:**

Have you tried varying the size of the filtered datasets to get a sense of the range at which is the proposed filtering approach the most effective?

**Reasons To Accept:**

1) The authors propose a simple method to filter a large corpora down by on average around 4x by removing sentences containing frequent tokens.
2) Reducing the required training data, and thus computation, of language model pretraining is an important area of research.
3) Authors demonstrated the efficacy of their approach on a comprehensive list of tasks and probes.
4) The authors intend to publish the datasets and code.
5) Idea and writing was easy to follow.

**Reasons To Reject:**

1) The explored design space of filtering algorithms was limited.

**Reproducibility:**

5: Could easily reproduce the results.

**Reviewer Confidence:**

2: Willing to defend my evaluation, but it is fairly likely that I missed some details, didn't understand some central points, or can't be sure about the novelty of the work.

**Typos Grammar Style And Presentation Improvements:**

Tables in general: I might consider using underline or bolding rather than red color to indicate emphasis for readability.

---

> ### Author Rebuttal · Authors · 2023-08-28
>
> Dear R1,
>
> We appreciate very much your positive opinion about our line of research. Indeed, it is crucial to find out: 1. that LM can also deliver good enough results for languages that do not count on very large pre-training datasets, 2. to understand what are the characteristics of optimal pre-training data sets maybe to make design guidelines to support low-resource languages or domains, and 3., last but not least, to be the first ones to set up the basis to empirically confirm that high frequency tokens distort the semantic space and harms similarity-based tasks.
>
> We have investigated other filtering methods and mostly learned from Fuster-Bagetto and Fresno (EMNLP 2022) results, as reported in section 2.  Note that since we are addressing most frequent words, only deleting them would produce rare, non-correct sentences that might not be suitable for the model that is assessing the probabilities of sequences of words.
>
> As for your question:
>
> 1. Have you tried varying the size of the filtered datasets to get a sense of the range at which is the proposed filtering approach the most effective?
>
> We have not done it in a systematic way, but only for Quechua, which gets a filtered dataset of 1M. Note that our ultimate goal is not the filtering method, but to understand the importance of having a word-frequency-balanced dataset when working with small datasets, and we take the 10M size as a minimal, critical amount for learning as demonstrated in Zhang et al. 2020. We leave as future work a more systematic comparison of the size of the filtered datasets for the other languages to better know the optimal range.

---

### Meta-Review · Area_Chair_u5XE · 2023-09-18

**Recommendation:** 4

**Metareview:**

This paper proposes to reduce the amount of data needed to achieve good language modeling perplexity and downstream task performance, especially that on semantic tasks, by removing sentences consisting entirely of high frequency tokens.

Reasons To Accept:
- The authors propose a simple method to filter a large corpora down by on average around 4x by removing sentences containing frequent tokens.
- Reducing the required training data, and thus computation, of language model pretraining is an important area of research.
- Authors demonstrated the efficacy of their approach on a comprehensive list of tasks and probes.
- The authors intend to publish the datasets and code.
- Idea and writing was easy to follow.
- This is an interesting idea that attempts to deal with power-law issues that may imply we as a field are brute-forcing in a non-intelligent way.

Reasons To Reject:
- The explored design space of filtering algorithms was limited.
- Need to evaluate the obtained results with full datasets used for obtaining 10M-freq. Not clear whether the baseline comparison is correct (whether were sampled same data)
- The suggested approach is quite straight-forward and there is no specification in what setups it could be beneficial

---

### Decision · Program_Chairs · 2023-10-07

**Decision:**

Accept-Findings

**Comment:**

This paper proposes to reduce the amount of data needed to achieve good language modeling perplexity and downstream task performance, especially that on semantic tasks, by removing sentences consisting entirely of high frequency tokens.

Reasons To Accept:
- The authors propose a simple method to filter a large corpora down by on average around 4x by removing sentences containing frequent tokens.
- Reducing the required training data, and thus computation, of language model pretraining is an important area of research.
- Authors demonstrated the efficacy of their approach on a comprehensive list of tasks and probes.
- The authors intend to publish the datasets and code.
- Idea and writing was easy to follow.
- This is an interesting idea that attempts to deal with power-law issues that may imply we as a field are brute-forcing in a non-intelligent way.

Reasons To Reject:
- The explored design space of filtering algorithms was limited.
- Need to evaluate the obtained results with full datasets used for obtaining 10M-freq. Not clear whether the baseline comparison is correct (whether were sampled same data)
- The suggested approach is quite straight-forward and there is no specification in what setups it could be beneficial